# IS THE FAIRNESS METRIC TRULY FAIR?

## ABSTRACT

Image classification is a fundamental task in computer vision that has been widely adopted in critical applications such as face recognition and medical imaging, drawing considerable attention to its predictive fairness. Some researchers have proposed various fairness metrics and pipelines to enhance the fairness of deep learning models. However, recent studies indicate that existing fairness evaluation specifications and metrics have inherent flaws, as they focus on low-dimensional inputs, such as numerical data, and overlook partial correlations between target and sensitive attributes, leading to some degree of mutual exclusivity. This raises the question: *Is the fairness metric truly fair?* Through in-depth analysis, experiments conclude that the fairness of deep models is closely related to attribute sampling and the interdependencies among attributes. In this work, we address this challenge by introducing a new specification based on dynamic perturbation for image classification models. Specifically, we introduce an Attribute Projection Perturbation Strategy (APPS) that moves beyond the constraints of directly statistical discrete predictions by mapping sensitive attributes that may influence task attributes onto the same dimension for evaluation. Building on this, a Projection Fairness Metric System is proposed to quantifing the upper and lower bounds of fairness perturbations, examining and evaluating the impact of mapped sensitive attributes on the fairness of task predictions from different perspectives. Additionally, we conducted systematic evaluation experiments and extensive discussions, demonstrating that the proposed evaluation specification offers better objectivity and interpretability compared to existing metrics, in 24 image classification models including CNN and ViT architectures. It is hoped that this work will promote the standardization of fairness evaluation pipeline and metrics.

## 1 INTRODUCTION

Image classification, an important branch of computer vision, has demonstrated strong capabilities in image processing and analysis, leading to widespread applications across multiple fields (Ren et al., 2015; Krizhevsky et al., 2012; Dosovitskiy et al., 2020). From facial recognition and medical diagnostic to autonomous driving, the capabilities of image classification models have significantly enhanced decision-making efficiency and accuracy in these domains, thereby making the issue of model fairness increasingly prominent (Buolamwini & Gebru, 2018).

Specifically, when models are extensively applied in real-life scenarios, specific groups may be systematically overlooked or misrepresented. For example, facial recognition systems exhibit significant accuracy disparities across different racial and gender groups (Crawford & Paglen, 2021; Raji & Buolamwini, 2019; Zhao et al., 2017); medical diagnostic models show varying accuracies for images from different hospitals due to differences in equipment type and quality, potentially leading to misdiagnoses or missed diagnoses that affect patients' health outcomes (Drukker et al., 2023; Zong et al., 2022); and in autonomous driving technology, if perception systems have biases in identifying pedestrians under different angles or lighting conditions, it could increase the risk of traffic accidents for certain groups (Pathiraja et al., 2024). For computer vision models, achieving comprehensive fairness across multiple attributes is crucial. Therefore, developing methods to evaluate and mitigate unfairness is essential to the safety and effectiveness of computer vision technologies.

To evaluate fairness of deep models more objectively, researchers have proposed various fairness metrics and evaluation pipelines in recent years. Early work primarily focused on statistical parity measures (Hardt et al., 2016), aiming to ensure consistency in prediction outcomes across different

groups. Subsequently, research shifted towards individual fairness metrics (Thomas et al., 2019), emphasizing the principle that similar individuals should receive similar treatment. Then, group fairness evaluation has become a focal point, evaluating model fairness by examining the performance of subgroups defined by different sensitive attributes (Mehrabi et al., 2021). However, these studies predominantly concentrate on low-dimensional inputs (such as numerical data) and often analyze traditional sensitive attributes (such as gender, age, and race) in isolation, neglecting partial correlations between targets and sensitive features. Moreover, due to the ambiguity in definition, the resulting metrics can be mutually exclusive in certain situations (Castelnovo et al., 2022).

In this work, we attempt to shed light on the black box of fairness evaluation by analyzing the relationships between multiple sensitive attributes and task attributes. Specifically, this paper aims to address two questions: *What factors limit the fairness evaluation*, and *how to objectively evaluate fairness?* To this end, we conduct extensive controlled variable experiments (including metrics, attributes and data distributions) to analyse the uncertainties involved in evaluating image classification models. Our in-depth research reveals that fairness evaluation methods can indirectly lead to incorrect results due to confused metric definitions, unstable multi-attribute results, and imbalanced test data labels and distributions, which also introduce additional subjective evaluation biases.

Based on these findings, we attempt to establish a new set of fairness evaluation specification that should satisfy two fundamental conditions: attribute sampling continuity and attribute correlation independence. Therefore, a new evaluation specification based on dynamic perturbation is propose to control the attributes during the evaluation process so that they meet the aforementioned conditions. Building on this, we introduce an Attribute Projection Perturbation Strategy (APPS) and a Projection Fairness Metric System, which projects discrete attribute labels into a continuous projection space for fairness evaluation. Experiment results demonstrate that existing label-based statistical metrics exhibit significant conflicts in distinguishing the effects of attributes themselves from those of related attributes. In contrast, the proposed method of evaluating dynamic perturbations of specific attributes within a unified projection space provides more consistent results and accurately identifies intrinsic correlations between attributes, thereby enhancing evaluation explainability and enabling more objective results through different mappers.

To sum up, the key contributions of our work can be listed as follows: **1)** The finding obtained from a compresive analysis indicates that ambiguous definitions, subjective labels, and coupling between attributes affect the evaluation of fairness, which will inspire more deep research in this area; **2)** We established a new fairness metric specification, stipulating the conditions that fairness evaluation should satisfy—uniform attribute sampling and attribute sampling independence, and proposed a metric system for approximating and quantifying the upper and lower bounds of fairness perturbations; **3)** Extensive experimental analyses of 24 image classification models indicate that evaluating model fairness in attribute projection space lead to better results than in discrete label space, which are closer to human cognition and more stable across different tasks.

## 2 RELATED WORK

In evaluating deep models' fairness, metrics can be roughly divided into group and individual fairness. In addition, the techniques of disentanglement representation are also involved in this paper.

**Group fairness** evaluation metrics first appeared with 'Demographic Parity' (DP) (Hardt et al., 2016), which requires that the positive prediction probabilities be the same across different groups, falling under the category of 'Group Parity'. Subsequently, Hardt et al. (2016) introduced 'Equal Opportunity' (EOpp) and 'Predictive Equality,' which require the model to maintain consistent true positive rates (TPR) and false positive rates (FPR) across subgroups, belonging to the 'Conditional Parity' category. In the same year, researchers proposed 'Equalized Odds' (EOdd) (Hardt et al., 2016), which requires the model to maintain consistent true positive rates and false positive rates across subgroups, further refining the concept of 'Conditional Parity'. Additionally, 'Treatment Equality' (TE) (Thomas et al., 2019) requires the model to maintain consistent false positive and false negative rates (FNR) across subgroups. Recently, 'Predictive Rate Parity' (PRP) ensures the independence of predicted positive individuals from subgroups, while 'Representation Disparity' (RD) (Hashimoto et al., 2018) ensures fairness by limiting representation disparity below a given threshold, falling under the category of 'Independence'. The development of these metrics reflects the ongoing efforts of researchers to refine the framework for evaluating model fairness.

**Individual fairness** evaluation metrics began with (Dwork et al., 2012), who introduced 'Individual Fairness', aiming to ensure that models treat similar individuals similarly in their predictions, regardless of their sensitive attributes. These metrics fall under the category of 'Similarity Fairness'. Subsequently, 'Metric Fairness' (Albarghouthi & Vinitsky, 2019) requiring deep neural networks to produce similar predictions for similar input samples, measuring similarity through distance metrics such as Euclidean distance or cosine similarity, thus further refining the concept of 'Similarity Fairness'. Counterfactual fairness (Kusner et al., 2017; Thomas et al., 2019) is based on the counterfactual causal relationship between sensitive attributes and prediction labels, requiring consistent prediction results for samples that are identical except for their sensitive attributes, falling under the category of 'Causal Fairness'. The development of these metrics reflects researchers' ongoing efforts to improve fairness evaluation at the individual level.

However, these methods focus only on the statistical results of single inferences and are susceptible to biases due to factors such as dataset and metric selection. We studied how to obtain comprehensive and objective results under an evaluation paradigm based on perturbations and attribute mapping. Contrary to estimates of individual similarity, we propose approximating the upper and lower bounds of task attribute decisions when sensitive attributes are changed as a measure, and introduce a new perspective for evaluating fairness.

**Disentangled representation** was introduced by Bengio et al. (2013), and it has demonstrated significant application value in multiple domains, including image generation/editing/translation, and multimodal applications. The core of disentangled learning lies in deeply understanding the latent factors of models and enhancing fine-grained controllability. Methods based on variational autoencoders (VAEs), such as $\beta$-VAE (Higgins et al., 2017), DIP-VAE (Kuriata et al., 2018), and FactorVAE (Kim & Mnih, 2018), achieve unsupervised disentanglement by directly constraining probability distributions. However, Locatello et al. (2019) pointed out the need for additional inductive biases. Subsequently, Burgess et al. (2018) proposed a method to incrementally increase the information capacity of latent variables, while Yang et al. (2019) used symmetry modeling based on group theory as an inductive bias. Additionally, research based on generative adversarial networks (GANs) has made progress, including the use of aging mutual information (Xu et al., 2019), self-supervised contrastive regularizers (Oord et al., 2018), and integration with other pretrained generative models. The rapid development of generative diffusion probability models (DPMs) has also promoted the learning of disentangled representations. These methods effectively extract high-dimensional information from images and map it to latent codes. Previously, disentanglement techniques have rarely been applied to fairness evaluation through continuous perturbations of image features. We present the first attempt to provide a new perspective for assessing model fairness.

## 3 INVESTIGATING THE INFLUENCING FACTORS OF FAIRNESS EVALUATION

Although previous work has studied fairness evaluation from perspectives such as symbolic definitions (Mitchell et al., 2021; 2018), metric comparisons (Castelnovo et al., 2022; Garg et al., 2020), and lifecycle analysis by (Du et al., 2020; Agarwal & Agarwal, 2022), most of this work is based on machine learning or tabular data. The fairness of deep models in the field of computer vision still lacks fine-grained analysis and understanding. We conducted a series of controlled variable experiments to evaluate the fairness of various image classification models, providing a comprehensive analysis of the issues present throughout the entire process of model fairness evaluation.

### 3.1 EXPERIMENTAL SETUP

To measure the fairness at different levels (confusion in metric definitions, instability in multi-attribute results, and imbalance in test data) in the current label-based fairness evaluation framework, we selected 3 mainstream metrics, 24 models, and 40 sensitive attributes for experimental analysis.

For the evaluation metrics, to minimize confusion due to inconsistencies in metric types, we adopted four metrics based on label-based fairness evaluation specification, namely Demographic Parity (DP), Equality of Opportunity(EOpp), Equalized Odds (EOdd), and Average Odds(AOdd) (Hardt et al., 2016). These fairness evaluation metrics are based on attribute label statistics and aim to measure the consistency of model performance across different groups. Specifically, DP focuses on the equality of overall positive prediction rates, while EOdd and EOpp address different aspects

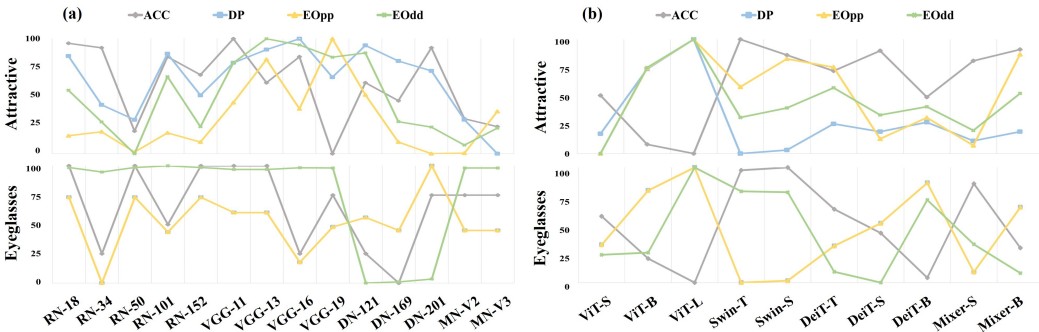

Figure 1: Comparison of fairness scores across different metrics. **(a)/(b):** Average scores of attribute 'Attractive' *(top)* and 'Eyeglasses' *(bottom)* with all sensitive attributes on CNN/Transformer models. 'RN', 'DN', and 'MN' represent ResNet, DenseNet, and MobileNet, respectively.

of prediction accuracy (the former requires equal false positive and false negative rates, while the latter requires equal true positive rates). AOdd provides a composite measure of differences in true positive and false positive rates. The formulas are provided in Appendix A.1.

For the tested attributes and models, we conducted large-scale experiments on CelebA (Liu et al., 2018), which contains 200,000 celebrity facial images, each labeled with 40 attributes, and a series of CNN-based (including ResNet, VGG, MobileNet) and Transformer-based (including ViT, Swin, DeiT) image classification models. We selected subjective attributes ('Attractive') and objective attributes ('Eyeglasses', 'Bald') as task attributes for training, and paired them with all other sensitive attributes for analysis. Then, we preprocessed the dataset to ensure sufficient sample sizes for each sensitive attribute to allow for reliable analysis. Then, we trained these models and evaluated their performance on sensitive attributes after each training phase. Finally, we examined multiple metrics at different evaluation levels and applied 0-1 reversal and min-max normalization to comprehensively assess the fairness performance of the models.

### 3.2 Observation

We make the following three main observations, which are consistent across different settings.

**The definitions of fairness evaluation metrics are partially conflicting and confusing.** In the process of fairness evaluation, the first step is to determine the evaluation metrics, however, current metrics are designed to meet different definitions of fairness. (Castelnovo et al., 2022) indicates that when the actual acceptance rates differ among groups, pursuing demographic parity (DP) may result in the failure to satisfy equal opportunity (EOpp). Similarly, striving for equalized odds (EOdd), which requires equal true positive and false negative rates, may compromise DP or EOpp. In Figure 1, we measure fairness using the average scores of models trained on task attributes across all sensitive attributes. Specifically, we observed that for the subjective task 'Attractive,' which involves more potential influencing factors, the scores and rankings provided by the metrics differ significantly between CNN and ViT architectures, as shown in Figure 1(top). In contrast, for the simple objective attribute 'Eyeglasses', all models exhibit high accuracy, with the DP and EOdd score curves nearly overlapping but differing significantly from EOdd, as illustrated in Figure 1(bottom). Even with the same specification, the fairness scores still depend on the metrics and definitions used.

**Selecting different sensitive attributes to evaluate fairness will result in different scores.** Given that fairness evaluation results are closely related to sensitive attributes, it is natural to ask whether selecting more sensitive attributes would lead to more objective evaluation results. In Figure 2(b), we trained the model on the task attribute 'Bald' and presented the fairness scores across all sensitive attributes using a radar chart. It is evident that there are significant differences between each metric, which further explains the reason for metric conflicts observed in Observation 1. Additionally, the metrics exhibit irregular responses across all sensitive attributes, with some even contradicting human cognition, such as attributing the unfairness of 'Bald' to unrelated sensitive attributes like 'Mouth_Slightly_Open', '5_o_Clock_Shadow' and 'Eyeglasses'. Therefore, when using existing fairness evaluation specifications, selecting incorrect or too many sensitive attributes can further exacerbate the unfairness of the evaluation process.

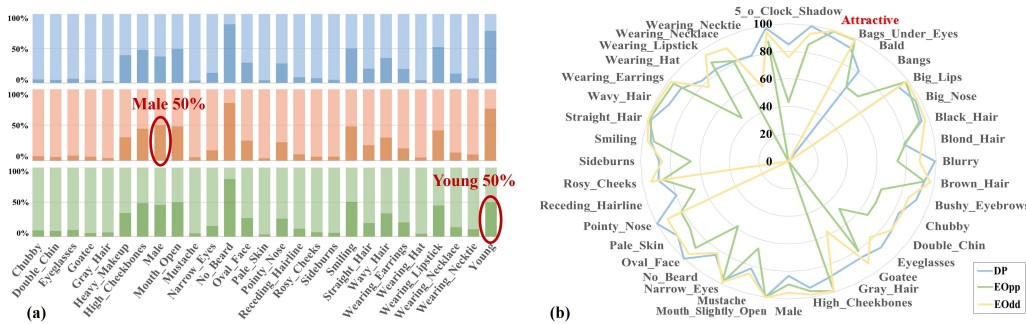

Figure 2: **(a):** Category distributions of all attributes in the test data: original*(blue)*, and uniform sampling of attributes 'Male'*(orange)* and 'Young'*(green)*. **(b):** Fairness scores of different sensitive attributes on the task attribute 'Attractive' in ResNet-50; lower scores indicate greater unfairness.

**Subjectivity in labeling and imbalance in test attributes result in inaccurate results.** After determining the appropriate evaluation metrics and the sensitive attributes to be tested, the inherent biases in the dataset will still significantly affect the results of the fairness evaluation. First, the subjectivity inherent in data annotation is itself a significant source of unfairness in computer vision models, primarily manifesting as annotator bias (Sheng et al., 2019; Berinsky et al., 2012), ambiguous labeling standards (Snow et al., 2008; Hovy et al., 2016), and cultural bias (Geva et al., 2019; Sap et al., 2019). These studies emphasize the need for clear, consistent labeling guidelines and a diverse pool of annotators to reduce subjectivity and enhance fairness. Consequently, the reliance of existing fairness evaluation specifications on labels can lead to unfairness in the evaluation process itself. Second, although there are sampling techniques that can appropriately alleviate the issue of long-tail distribution in data labels, achieving uniform sampling across all sensitive attributes in the test data remains a significant challenge. To further validate this point, we calculated the proportions of each of the 40 attributes in the test dataset (first row of Figure 2(a)) and obtained two new datasets through uniform sampling for the attributes 'Male' and 'Young' (second and third rows of Figure 2(a)). It is evident that the distributions of other attributes remained largely unchanged. We re-measured the model's scores on these metrics using the new test sets and found that the variations in results were within 3% which demonstrates that balancing a single attribute alone cannot rectify the inherent unfairness of the dataset.

Our analysis highlights the significant inherent unfairness of the metrics under existing fairness evaluation specifications during the evaluation process. Next, we will address how to systematically correct these issues in the evaluation process to achieve truly fair and objective assessment results.

## 4 FAIENESS EVALUATION SPECIFICATION

Based on the above observations, the evaluation methods for model fairness can indirectly lead to incorrect evaluation results due to many reasons such as ambiguous metric definitions, unstable results across multiple attributes, and imbalanced testing data labels and distributions. Meanwhile, these issues can also introduce additional subjective evaluation biases. Therefore, to objectively and systematically evaluate the fairness performance of image classification models, we propose a new evaluation specification: the degree to which task attributes are influenced by sensitive attributes. This concept aims to measure the model's sensitivity to changes in non-critical features when predicting task attribution. Due to the inability to directly compute the influence in high-dimensional feature spaces, inspired by Moosavi-Dezfooli et al. (2016), we project various sensitive features into a unified, continuous, and controllable space through specialized mappers to approximate their perturbation bounds, thereby evaluating model's fairness, as shown in Figure 3.

Formally, for a given classifier, we define the fairness bounds of each attribute in the D-dimensional projection space as the minimal perturbation $\hat{a}_d$ sufficient to alter the estimated label $\hat{l}(U(z))$:

$$\Delta(z_d; \hat{l}) := \min_{\hat{a}_d} \|\hat{a}_d\|_1 \text{ subject to } \hat{l}(U(z \oplus \hat{a}_d)) \neq \hat{l}(U(z)), \tag{1}$$

Figure 3: Overview of the proposed evaluation specification. In the attribute space *(right)*, test datas are divided into different levels according to discrete labels, from coarse to fine: entirety, group, and individual. Then, we introduce different mappers and perturbation strategy to switch the evaluation perspective to the projection space *(left)*. By controlling the projection factor $z$ to approximate the fairness boundary $\hat{a}_d$, the subjective labels will be unified to a continuous dimension. Finally, the proposed metric system corresponding to each level in attribute space will be used to objectively evaluate the fairness of models.

where $z$ is the projection factor obtained through the mapper, which controls the continuous changes after sensitive attributes are mapped to a specific space; $\hat{l}(U(z))$ denotes the predicted label of task attribute; $\oplus$ indicates that the perturbation $\hat{a}_d$ is only added to the $d$-th dimension of $z$, leaving the rest unchanged. We denote $\Delta(z_d; \hat{l})$ as the fairness boundary of $\hat{l}$ at point $z_d$, further distinguished into upper bound $\hat{a}_d^+$ and lower bound $\hat{a}_d^-$ according to the direction of $\hat{a}_d$.

It is important to note that the evaluation methods proposed above do not alter the traditional definition of fairness, which requires the model to maintain consistency and impartiality when predicting task attributes, avoiding any form of bias or discriminatory behavior.

## 5 PROJECTION FAIRNESS EVALUATION FRAMEWORK

We revisited the perspective of fairness evaluation and introduced a projection fairness evaluation framework for visual deep models, achieving unified mapping of various sensitive attributes and their values, and approximating the fairness boundary to evaluate the fairness of image classification models at different levels. Specifically, within this framework, we proposed an Attribute Projection Perturbation Strategy (APPS) and a set of metrics related to fairness definition, providing an objective method for evaluating the fairness of visual deep models in classification tasks.

### 5.1 ATTRIBUTE PROJECTION PERTURBATION STRATEGY

To map the sensitive attributes that may influence task attributes onto the same dimension and standard, we define a mapping function $f : \mathbb{R}^{C \times H \times W} \to \mathbb{R}^D$, which maps an image $I \in \mathbb{R}^{C \times H \times W}$ to a perturbation space projection factor $\mathbf{z} \in \mathbb{R}^D$. For the factor $z$, we will evaluate the overall fairness of a model by calculating its impact on fairness for each dimension $d$. This mapping function can be adapted according to different perspectives of fairness evaluation. One potential solution is to use the disentanglement capabilities of autoregressive or generative models as mappers, generating a more comprehensive latent code that covers sensitive attributes within the dataset. This approach can mitigate the limitations of manually specifying sensitive attributes.

After determining the mapping function $f$, a dynamic perturbation iterative process can be employed to estimate the fairness boundary $\Delta(z_d; \hat{l})$. We use the model's predictions after changes in sensitive attributes as soft labels for evaluation, approximating fine-grained continuous variations of single attributes to estimate the fairness boundary. Specifically, in each iteration, we first apply positive and negative perturbations of magnitude $\sigma$ to the projection factor $z$ obtained from the mapping of sensitive attributes, computing the perturbation of the $d$-th attribute of the encoding $z \in \mathbb{R}^D$ as $\hat{a}_d^{\pm} = \mu_d \pm \sigma_d \cdot \frac{m}{M}$, where $m$ and $M$ denote the current iteration count and its upper limit, while $\mu$ and $\sigma$ denote the perturbation center and the unit (both learned through an autoregressive model or sampled from the projection range), respectively. The projection perturbation algorithm for each classifier is summarized in Algorithm 1. The algorithm stops when $U(z \oplus \hat{a}_d)$ changes the prediction

---

**Algorithm 1** Projection Perturbation Algorithm

---

1: **Input:** initial value $\mu \in \mathbb{R}^D$, perturbation unit $\sigma \in \mathbb{R}^D$, classifier $f$, decoder $U$, iterations $M$
2: **Output:** Perturbation $\hat{a}^+ \in \mathbb{R}^D$, $\hat{a}^- \in \mathbb{R}^D$
3: Initialize $z \leftarrow \mu$, $\hat{a}^+ \leftarrow 0$
4: **for** each attribute $d$ in $D$ **do**
5:    $m \leftarrow 0$
6:    **while** $f(U(z \oplus \hat{a}_d^+)) = f(U(z))$ **and** $m < M$ **do**
7:       $\hat{a}_d^+ = \mu_d + \sigma_d \cdot \frac{m}{M}; m \leftarrow m + 1;$
8:    **end while**
9: **end for**
10: Repeat steps 3-8 to calculate the negative minimal perturbation $\hat{a}^-$
11: **Return** $\hat{a}^+$, $\hat{a}^-$

---

results of the classifier, and we can approximately determine the upper bound $\hat{a}_d^+$ and lower bound $\hat{a}_d^-$ of the influence of each sensitive attribute's projection factor on the task prediction.

## 5.2 PROJECTION FAIRNESS METRIC SYSTEM

Most previous work on fairness evaluation has been based on label statistics, utilizing discrete labels of samples to divide data into different levels of granularity (Figure 3 right). Group and individual fairness compute probability statistics between a single attribute and task labels across different value intervals, and between other attributes' similarity and task labels when the single attribute is fixed. However, these approaches suffer from the three issues discussed in Section 3.2. For the mapped attribute space (Figure 3 left), modifying the projection factor $z$ allows for the precise alteration of one or more attributes, achieving more accurate, finer-grained, and continuously controllable data division results (including both types of divisions mentioned above). Based on this, analyzing the fairness boundaries $\hat{a}$ of different attributes allows for evaluating different properties of fairness within the same dimension. Below, we provide some possible evaluation metrics:

**Tolerance.** The maximum extent to which perturbations applied to the projection factors of sensitive attributes can be accepted in terms of task attributes, which indicates the range of similarity in sensitive attribute values that the model tolerates. The Tolerance score $Tol$ of all sensitive attributes is quantified by $\frac{1}{D} \sum_{d=1}^{D} (\hat{a}_d^+ - \hat{a}_d^-)/\sigma_d$, where $D$ denotes the dimension of the sensitive attributes, $\hat{a}$ denotes the perturbation boundary, and $\sigma$ denotes the perturbation unit. For each sensitive attribute, a higher score indicates a smaller impact on the fairness of task attributes, similar to the effect of similarity on prediction results in *individual fairness* evaluations (which is difficult to calculate precisely in the attribute space). Evaluating in the projection space circumvents the constraints of sampling the attributes themselves, thus allowing the similarity of attributes to be quantified.

**Deviation.** The range of perturbations applied to the projection factors of sensitive attributes relative to the perturbation center, which indicates the model's preference for the value intervals of sensitive attributes. The score $Dev$ of all sensitive attributes is quantified by $\frac{1}{D} \sum_{d=1}^{D} \left| (\hat{a}_d^+ + \hat{a}_d^-)/2 - \mu_d \right|$, where $D$ denotes the dimension of the sensitive attributes, $\hat{a}$ denotes the perturbation boundary, and $\mu$ denotes the perturbation center. For each sensitive attribute, a higher score indicates a greater deviation from the perturbation center, similar to the bias towards specific value groups in *group fairness* evaluations. Evaluating in the projection space circumvents the constraints of discrete labels, allowing for the computation of continuous values of model bias within the range of sensitive attribute variations, rather than being limited to given labels.

**Coupling.** Evaluate the degree of correlation between task attributes and all sensitive attributes within the context of the dataset. The score $Cou$ of all sensitive attributes in projection space is quantified by $-\sum_{d=1}^{D} P_d \log P_d$, where $P_d = 1 - Tol_d / \max_{d=1}^{D} Tol_d$ denotes the 'probability' of correlation between sensitive attributes and task attributes, inspired by Eastwood & Williams (2018), This metric is applicable when using disentanglement models, such as autoregressive models, as mappers. It allows one to avoid the constraints imposed by the manual selection of sensitive attributes being measured. If the influence of the sensitive attribute's projection factor $z$ on the task attribute is significantly imbalanced, the score will be higher.

Table 1: Fairness scores of different model on the CelebA dataset for the task attributes 'Attractive' and 'Eyeglasses'. Acc, DP, EOpp, and EOdd represent accuracy, demographic parity, equality of opportunity, and equalized odds, respectively. Tol and Dev are the metrics tolerance and deviation proposed in this paper. Results are averaged over 40 sensitive attributes.

| | Model | Attractive | | | | | | Eyeglasses | | | | | |
|---|---|---|---|---|---|---|---|---|---|---|---|---|---|
| | | Acc | DP | EOpp | EOdd | Tol | Dev | Acc | DP | EOpp | EOdd | Tol | Dev |
| CNN | ResNet-50 | 80.98 | 30.98 | 21.02 | 28.15 | 93.77 | 46.90 | **99.69** | 10.01 | 10.01 | **7.61** | 99.30 | 49.38 |
| | ResNet-152 | 81.48 | 30.53 | 20.59 | 27.26 | 93.86 | 46.86 | **99.69** | 10.01 | 10.01 | **7.61** | 99.30 | 49.39 |
| | VGG-13 | 81.41 | **29.69** | 17.08 | **24.35** | 93.60 | 46.77 | **99.69** | 10.03 | 10.03 | **7.61** | 99.25 | 49.35 |
| | VGG-19 | 80.78 | 30.20 | **16.20** | 24.96 | 93.97 | 47.03 | 99.65 | 10.05 | 10.05 | **7.61** | 99.29 | 49.36 |
| | DenseNet-169 | 81.25 | 29.90 | 20.59 | 27.09 | **94.18** | **47.04** | 99.53 | 10.06 | 10.06 | 7.73 | **99.34** | **49.43** |
| | DenseNet-201 | **81.72** | 30.08 | 21.09 | 27.27 | 93.84 | 46.90 | 99.65 | **9.97** | **9.97** | 7.73 | 99.28 | 49.37 |
| | MobileNet-V2 | 81.09 | 30.98 | 21.06 | 27.87 | 93.06 | 46.50 | 99.65 | 10.06 | 10.06 | **7.61** | 99.19 | 49.30 |
| Trans. | ViT-S | 74.92 | 31.42 | 23.15 | 32.68 | **95.95** | **47.78** | 96.95 | 7.07 | 7.07 | 16.07 | 99.21 | 49.54 |
| | ViT-B | 70.55 | **26.26** | 18.88 | **26.18** | 94.66 | 47.36 | 94.61 | **2.58** | **2.58** | 15.83 | **99.61** | **49.80** |
| | Swin-T | **79.92** | 33.01 | 19.82 | 29.95 | 94.00 | 46.96 | 99.49 | 10.19 | 10.19 | **8.85** | 99.29 | 49.42 |
| | Swin-S | 78.52 | 32.72 | **18.42** | 29.23 | 94.08 | 47.00 | **99.65** | 10.04 | 10.04 | 8.96 | 99.33 | 49.45 |
| | DeiT-T | 77.11 | 30.63 | 18.84 | 27.71 | 94.54 | 47.15 | 97.34 | 7.16 | 7.16 | 18.00 | 99.39 | 49.67 |
| | DeiT-S | 78.91 | 31.26 | 22.41 | 29.76 | 94.65 | 47.33 | 96.02 | 5.29 | 5.29 | 19.23 | 99.47 | 49.72 |
| | Mixer-S | 78.01 | 31.99 | 22.75 | 30.93 | 95.37 | 47.49 | 98.75 | 9.32 | 9.32 | 14.88 | 99.41 | 49.62 |

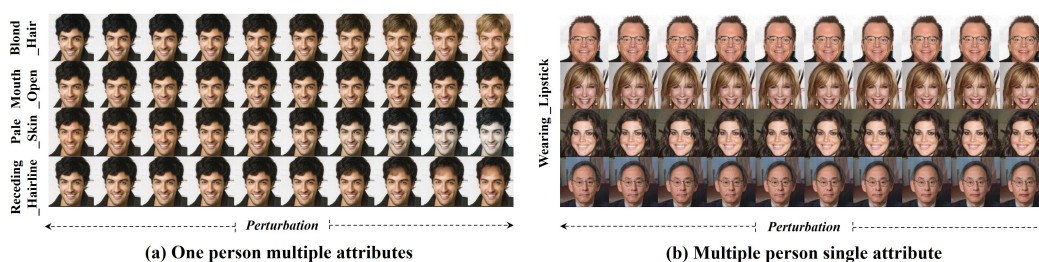

(a) One person multiple attributes       (b) Multiple person single attribute

Figure 4: Visualization of attribute projection perturbation in face reconstruction.

## 6 EXPERIMENT

**Models.** In order to comprehensively evaluate the current mainstream visual deep models and explore the impact of model architecture, depth, and parameter count on fairness, we selected a total of 14 CNN architectures and 10 Vision Transformer (ViT) architectures for the experiment. Specifically, the CNN architectures include ResNet (He et al., 2016), VGG (Simonyan & Zisserman, 2014), DenseNet (Huang et al., 2017), MobileNet (Howard et al., 2017), while the Visition Transformers architectures include ViT (Dosovitskiy et al., 2020), DeiT (Touvron et al., 2021), Swin (Liu et al., 2021), and Mixer (Tolstikhin et al., 2021). Within each series of models, we also chose several networks with different depths. All experiments use the same data augmentations provided by timm (Wightman, 2019), with images size uniformly scaled to 224×224, AdamW optimizer with weight decay of 0.05, drop-path rate of 0.1, gradient clipping norm of 1.0, and a cosine annealing learning rate schedule with a linear warm-up. Automatic mixed precision training strategy is adopted to accelerate training. All other training settings, including batch size, learning rate, warm-up periods, weight initialization strategies, and so on, are kept consistent across all comparative experiments.

**Evaluation.** During the mapping process of sensitive attributes, we opted for commonly used decoupling methods, specifically $\beta$-VAE and conditional GAN as the mappers. The visualization of attribute projection perturbation is shown in Figure 4. Our parameter settings are based on (Higgins et al., 2016; Burgess et al., 2018), with learning rate of $1e^{-4}$, $\beta$ of 10, and the image size is scaled to 128×128. For dynamic perturbation evaluation, we set the maximum number of iterations $m$ to 20. Our qualitative analysis revealed that larger perturbations of the projection factors tend to reduce the accuracy of image reconstruction. In the comparative experiments, the values for DP, EOdd, EOpp, Tol and Dev metrics range from 0 to 1, where a smaller value indicates better performance, while the Tol metric is preferable with larger values. Therefore, after normalizing the computed metrics, all indicators are scaled to a range of 0 to 100, where a higher value indicates better performance.

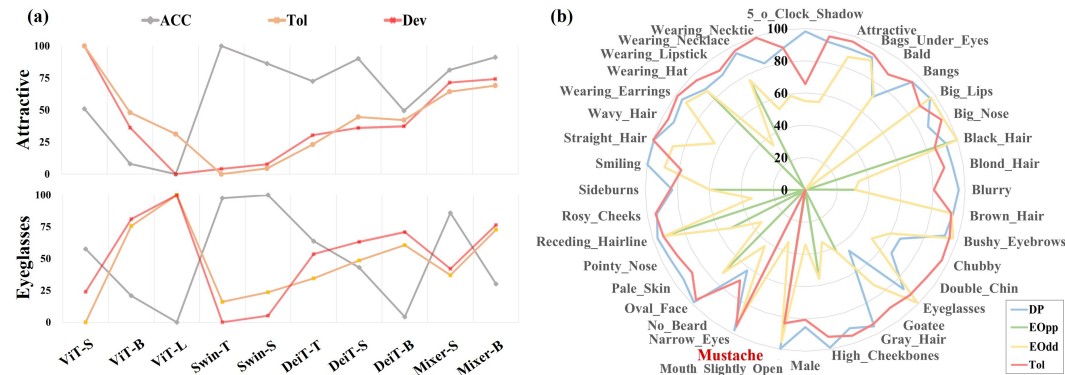

Figure 5: **(a):** Average scores of task attributes 'Attractive'*(top)* / 'Eyeglasses'*(bottom)* with all sensitive attributes on CNN models. **(b):** Normalized fairness scores of different sensitive attributes on the task attribute 'Mustache' in ResNet-50; lower scores indicate greater unfairness.

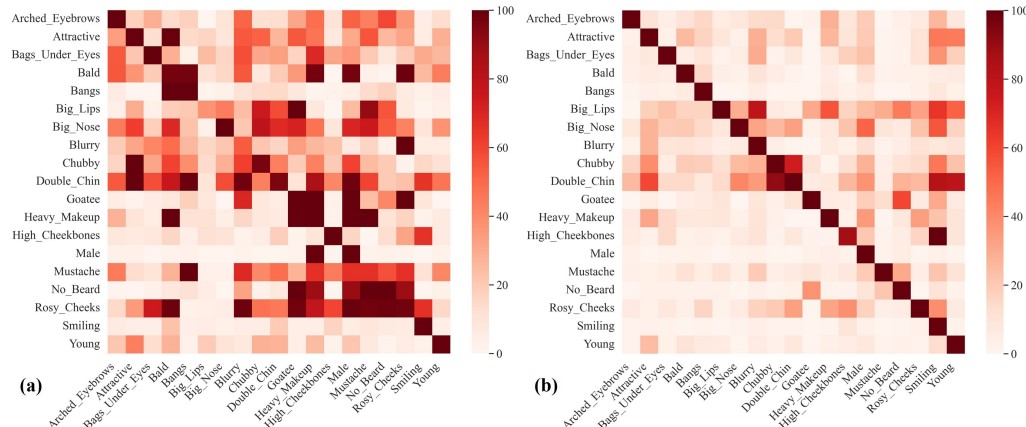

Figure 6: Correlation between attributes evaluated by EOpp**(a)** and Tol**(b)**. The strength of the correlation is represented by the fairness scores when attributes are used as task*(rows)* and sensitive*(columns)* attribute; deeper colors indicate stronger correlations.

## 6.1 RESULT AND ANALYSIS

We comprehensively evaluated the relationship between task attributes and multiple sensitive attributes across 15 image classification models, including CNN and ViT architectures, using a variety of metrics from different perspectives. The scores for these metrics are presented in Table 1. To better illustrate the advantages of the proposed evaluation framework, we will analyze the experimental results in terms of Stability, Interpretability, and Independence.

**Stability.** As shown in Figure 1 and Figure 5(a), we selected both subjective ('Attractive') and objective ('Eyeglasses') attributes as task attributes, using the average metrics across all sensitive attributes to evaluate the models. A comparative analysis of these line charts reveals significant discrepancies in the results of metrics $DP$, $EOdd$, and $EOpp$ within the same model, with even more pronounced differences between subjective and objective attributes. In contrast, the analysis of Figure 5(a) shows that metrics $Tol$ and $Dev$, which unify the evaluation perspective within the same projection space, yield more consistent results within the same model, leading to similar rankings.

**Interpretability.** The radar chart in Figure 5(b) presents fairness scores for ResNet-50 between all sensitive attributes and task attribute 'Mustache'. A score closer to the center of the chart indicates a greater impact of unfairness from sensitive attributes. The result shows that metrics $DP$, $EOdd$, and $EOpp$ respond significantly to most sensitive attributes but fail to identify attributes with inherent correlations in the images. Even worse, these metrics exhibit significant distributional biases in attribute labels; for instance, many task attributes show generally low scores for attributes 'Chubby' and 'Double_Chin' (in the lower right corner), despite a lack of actual correlation, likely due to severe distributional biases in the test dataset, as seen in Figure 2. In contrast, metric $Tol$ (same

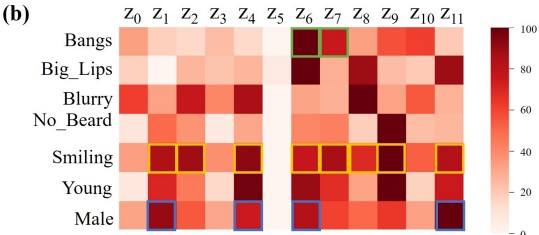

Figure 7: Correlation between disentangled projection factors $z$ of mapper $\beta$-VAE and task attributes. For ease of comparison, each color of the box represents a different attribute. **(a):** Qualitative relationship between projection factors and attributes in reconstructed images; '✓' indicates that perturbing $z_d$ changes the corresponding attribute. **(b):** Correlation between the task 'Attractive' and projection factors in ResNet-50, where the heatmap visualizes the scores of metric $Tol$.

trend as $Dev$) accurately identify sensitive attributes with meaningful image-level correlations to the task attributes, such as '5_o_Clock_Shadow', 'Sideburns', 'No_Beard', and 'Male'.

**Independence.** The heat map in Figure 6 displays fairness scores of sensitive attributes for ResNet-50 on different task attributes from the CelebA dataset, with scores reflecting the degree of mutual influence between attributes. The results reveals that the metric $EOpp$ is difficult to distinguish the influence of the attribute itself from other related attributes, resulting in score conflicts among metrics. In contrast, the metric $Tol$ effectively distinguish between the impact of the attribute in question and other related attributes. The results of other metrics (e.g. $DP$, $EOdd$, $Dev$) are shown in Appendix A.3. This improvement is attributed to the dynamic perturbation specific to each attribute, which significantly reduces reliance on hard labels and data distribution, allowing the evaluation to focus more on the intrinsic characteristics of the attributes.

## 6.2 EXTAND TO DISENTANGLEMENT

The experiments above have demonstrated that single-attribute perturbation based on condition GAN can provide an objective fairness evaluation results. Furthermore, when mapper can project combinations of multiple potential attributes in the data scene, the proposed specification can effectively extend to scenarios involving multiple intersecting attributes. Figure 7(a) presents the qual-

Table 2: Scores of metric $Col$ on different task attributes with mapper $\beta$-VAE.

| Model | Attractive | Smiling | Mustache |
|---|---|---|---|
| ResNet-50 | 3.98 | 5.80 | **2.68** |
| ViT-S | 5.60 | 6.09 | **2.71** |

itative analysis results of projection factors $z$ disentangled by $\beta$-VAE, which is associated with the results in Figure 7(b). The former shows that projection factor $z_d$ (each column) responds to multiple task attributes. We invited 10 experts to evaluate the associated attributes. The latter reflects the correlation between task attributes (each row) and each dimension factor $z$ evaluated on ResNet-50. Experiment shows that factors with a significant impact on task attributes can correctly control the associated attributes after reconstruction (such as Bang, Smile, Gender), which to some extent demonstrates that the relationship between metric $Tol$'s response to $z$ is consistent with human cognitive intuition. The scores of the metric $Cou$ in Table 2 also demonstrate that the objective task attribute 'Mustache' has lower coupling of sensitivity attribute compared to subjective tasks.

## 7 CONCLUSION

This work aims to address the lack of objectivity in fairness evaluation for image classification tasks. First, experiments analyzed the shortcomings of existing fairness evaluation frameworks at various stages of visual tasks. Next, a new evaluation specification based on dynamic perturbation is proposed to mitigate these issues. Building on this, we introduce an Attribute Projection Perturbation Strategy (APPS) and a Projection Fairness Metric System to map different attributes to the same dimension and evaluate their fairness impact on task predictions from multiple perspectives. Comprehensive experimental validation demonstrates that the proposed evaluation specification achieves greater objectivity and interpretability, aligning more closely with human understanding compared to existing evaluation methods. In the future, we plan to extend this work to other tasks in computer vision to establish a unified benchmark for fairness evaluation.

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

# A APPENDIX

In the supplements, we present the definitions and formulas for fairness metrics based on label statistics (A.1), followed by an analysis of stability (A.2), independence (A.3), interpretability (A.4), and attribute reconstruction (A.5) within the projection fairness evaluation framework.

## A.1 STATISTICAL FAIRNESS METRICS

The definitions in this section focus on a predicted outcome $R$ for various distributions of subjects, where $Y$ denotes the ground truth of task attribute and $A$ denotes the ground truth of the secondary attribute to be evaluated. These variables all satisfy the condition where "+" represents the correct label, and "-" represents any other incorrect label. They are the simplest and most intuitive notions of fairness. Below are the fairness metrics compared and analyzed in the main text:

**Demographic Parity**, also referred to as statistical parity, acceptance rate parity. A classifier satisfies this definition if the subjects in the protected and unprotected groups have equal probability of being assigned to the positive predicted class. This is, if the following formula is satisfied:

$$DP = |P(R = + \mid A = +) - P(R = + \mid A = -)|, \tag{2}$$

**Equal Opportunity**, also referred to as false negative error rate balance. A classifier satisfies this definition if the subjects in the protected and unprotected groups have equal FNR. This is, if the following formula is satisfied:

$$EOpp = |P(R = - \mid Y = +, A = +) - P(R = - \mid Y = +, A = -)|, \tag{3}$$

**Equalized odds**, also referred to as conditional procedure accuracy equality and disparate mistreatment. A classifier satisfies this definition if the subjects in the protected and unprotected groups have equal TPR and equal FPR, satisfying the formula:

$$EOdd = Max\big(P(R = + \mid Y = \pm, A = +), \quad P(R = + \mid Y = \pm, A = -)\big), \tag{4}$$

**Adversarial Odds**, also known as average equality of opportunity, aims to ensure that the predictions of a classifier are independent of the protected attribute while maintaining high accuracy. A classifier achieves AOdd if it satisfies the following conditions:

$$AOdd = Mean\big(P(R = + \mid Y = \pm, A = +), \quad P(R = + \mid Y = \pm, A = -)\big), \tag{5}$$

It can be seen that these result-based statistical metrics assess fairness by filtering predictions that align with a specific fairness definition, leading to a strong dependence on the data and labels. Due to the similar fairness constraints between $AOdd$ and $EOdd$, this study only selects metrics $DP$, $EOpp$ and $EOdd$ for analysis.

## A.2 EVALUATION STABILITY ANALYSIS

This section serves as an extension of the stability of evaluation part in the main text. As shown in Table 3, the fairness results of 10 models (including both CNN and Transformer architectures) outside the main text demonstrate the average evaluation scores for task attributes 'Attractive' and 'Eyeglasses' across all sensitive attributes. For CNN architectures, it can be observed that existing fairness metrics do not effectively distinguish between different models, with some metrics even causing overlapping statistical results. For transformers architectures, due to significant accuracy differences on the experimental datasets, all statistical fairness metrics exhibit considerable bias. In contrast, the proposed evaluation framework, by constructing a multidimensional measurement system, reveals significant differences among models across different metrics. Additionally, dynamic sampling perturbation evaluation not only considers single-sample estimates but also better mitigates the impact of insufficient accuracy on fairness testing. The results in Figure 3 illustrate the stability of the dynamic perturbation evaluation method more intuitively. The main text presents the results of the transformers-based models; here, we report the evaluation results for task attributes 'Attractive' (first row) and 'Eyeglasses' (second row) on CNN architectures, covering both subjective and objective task attributes. Scores of metrics $DP$, $EOpp$, and $EOdd$ show confusing fluctuations when evaluating subjective attributes, and partially repeated results when evaluating objective attributes. Under the proposed dynamic perturbation evaluation strategy, different metrics, including $Tol$ and $Dev$, maintain relatively stable evaluation results across different attributes and models.

Table 3: Fairness scores of additional models on the CelebA dataset for the task attributes 'Attractive' and 'Eyeglasses'. Acc, DP, EOpp, and EOdd represent accuracy, demographic parity, equality of opportunity, and equalized odds, respectively. Tol and Dev are the metrics tolerance and deviation proposed in this paper. Our results are averaged over 40 sensitive attributes.

| | Model | Attractive | | | | | | Eyeglasses | | | | | |
|---|---|---|---|---|---|---|---|---|---|---|---|---|---|
| | | Acc | DP | Eopp | Eodd | Tol | Dev | Acc | DP | Eopp | Eodd | Tol | Dev |
| CNN | ResNet-18 | 81.76 | 29.80 | 20.32 | 26.05 | 93.73 | 46.83 | 99.69 | 10.01 | 10.01 | 7.61 | 99.27 | 49.36 |
| | ResNet-34 | 81.72 | 30.71 | 20.15 | 27.10 | 93.92 | 46.91 | 99.57 | 10.13 | 10.13 | 7.61 | 99.27 | 49.37 |
| | ResNet-101 | 81.64 | 29.76 | 20.20 | 25.60 | 93.65 | 46.83 | 99.61 | 10.06 | 10.06 | 7.61 | 99.25 | 49.35 |
| | VGG-11 | 81.80 | 29.93 | 18.91 | 25.14 | 94.03 | 47.01 | 99.69 | 10.03 | 10.03 | 7.61 | 99.25 | 49.35 |
| | VGG-16 | 81.64 | 29.48 | 19.17 | 24.55 | 93.29 | 46.64 | 99.57 | 10.10 | 10.10 | 7.61 | 99.26 | 49.34 |
| | DenseNet-121 | 81.41 | 29.61 | 18.58 | 24.82 | 93.57 | 46.71 | 99.57 | 10.04 | 10.04 | 7.73 | 99.28 | 49.38 |
| | MobileNet-V3 | 81.02 | 31.61 | 19.29 | 27.30 | 94.10 | 47.01 | 99.65 | 10.06 | 10.06 | 7.61 | 99.19 | 49.30 |
| Trans. | ViT-L | 69.73 | 23.90 | 17.45 | 24.05 | 93.92 | 47.22 | 93.28 | 0.67 | 0.67 | 6.13 | 99.74 | 49.89 |
| | DeiT-B | 74.77 | 30.51 | 21.36 | 29.14 | 94.68 | 47.31 | 93.55 | 1.95 | 1.95 | 9.84 | 99.53 | 49.75 |
| | Mixer-B | 79.02 | 31.26 | 18.20 | 28.13 | 95.43 | 47.53 | 95.20 | 3.97 | 3.97 | 18.15 | 99.60 | 49.78 |

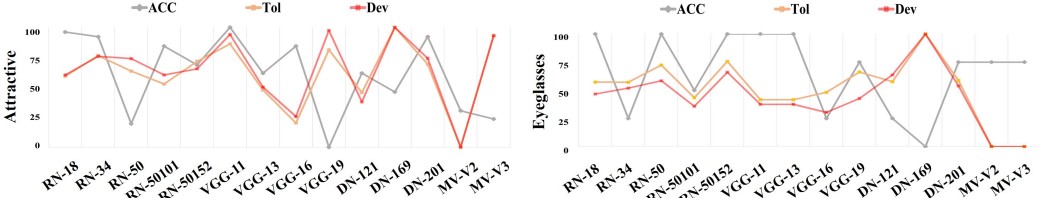

Figure 8: Normalized scores of fairness metric evaluations for different CNN models on the task attributes 'Attractive' and 'Eyeglasses'. 'RN', 'DN', and 'MN' represent ResNet, DenseNet, and MobileNet, respectively.

### A.3 EVALUATION INDEPENDENCE ANALYSIS

This section serves as an extension of the independence of evaluation part in the main text. The independence analysis experiment was conducted using the ResNet50 model on the CelebA dataset to explore its fairness scores across different task attributes and secondary attributes. These scores can be considered as indicators of the degree of interaction between different attributes. As shown in the heatmap in Figure 6, our proposed dynamic perturbation-based evaluation metrics (Tol and Dev) demonstrate higher independence compared to the existing label-statistics-based methods (DP, EOdd, and EOpp). In addition to the EOdd and Tol metric evaluation results reported in the main text, Figure 9 presents the evaluation results for the remaining metrics $DP$, $EOPP$, and $Dev$. Comparative analysis of the experimental results indicates that the traditional metrics $DP$, $EOdd$, and $EOpp$ struggle to effectively distinguish the impact of the primary attribute from other related attributes, leading to conflicting results among these metrics. Additionally, in certain instances, the metrics $EOdd$ and $EOpp$ do not effectively capture the intrinsic characteristics of the attributes. Conversely, metrics $Tol$ and $Dev$ demonstrate a greater capacity for discerning the differential impact between the primary attribute and its associated secondary attributes. This enhancement stems from the implementation of the dynamic perturbation strategy, which minimizes reliance on rigid labels and data distribution during evaluation, thereby enabling a more focused analysis on the inherent properties of the attributes.

### A.4 EVALUATE INTERPRETABILITY ANALYSIS

This section serves as an extension of the interpretability of evaluation part in the main text. Figure 10 and 11 illustrate the fairness scores of the ResNet50 model across all secondary attributes for a single task attribute in the CelebA dataset. Attributes closer to the center indicate a greater degree of unfairness. Comparative analysis reveals that metrics $DP$, $EOdd$, and $EOpp$ exhibit significant responses to most secondary attributes but fail to identify attributes with semantic relevance within the images. In contrast, the metrics $Tol$ and $Dev$ effectively identify secondary attributes that have a semantically meaningful relationship with the task attribute. For example, the 'Heavy_Makeup' attribute shows poorer fairness scores for 'Wearing_Lipstick', Attractive, and

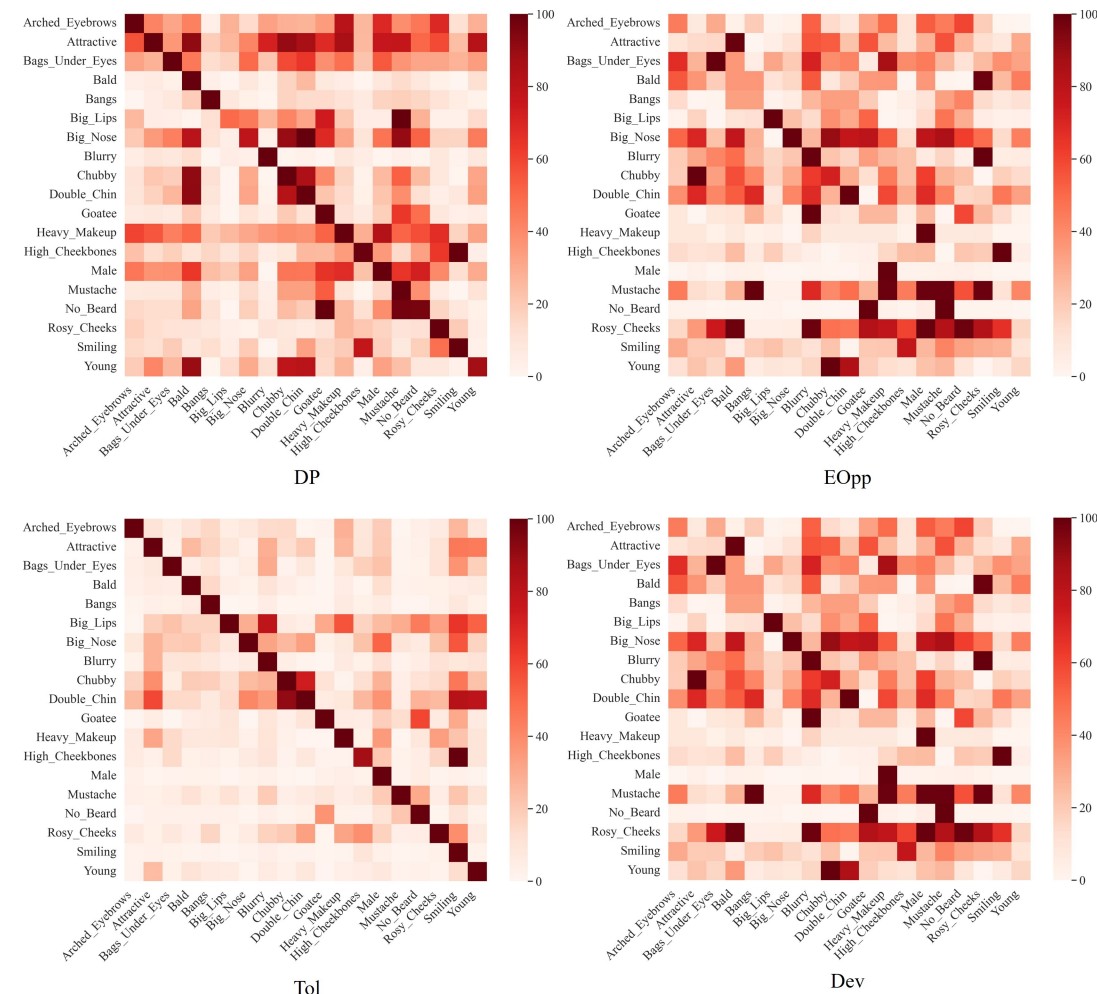

Figure 9: The correlation between attributes evaluated by $DP$, $EOpp$, $Tol$ and $Dev$. The values represent normalized scores when attribute are considered as task attributes and sensitive attributes.

'Rosy_Cheeks'. Similarly, our evaluation indicates that the Bald attribute is strongly associated with 'Wearing_Hat', 'Wavy_Hair', 'Receding_Hairline', 'Male', and 'Bangs'. Moreover, for the attribute 'High_Cheekbones', there is a strong association with the smiling attribute, aligning well with human intuitions about the relationships of attributes.

## A.5 VISUALIZATION OF ATTRIBUTE PERTURBATION

We examined each attribute after perturbing to ensure that the attributes reconstructed with fidelity could be controlled by individual projection factors. For each attribute, we applied dynamic sampling perturbations to a controllable projection factors and reconstruct the new attributes back to the original images. As shown in Figure 12, the results confirm our expectations: continuous sampling of a single projection factor in the projection space can independently control a single attribute in the image. For ease of comparison and to observe the independence of control, the same test dataset was used for each attribute's display. Larger areas of projection, such as hair and skin color, are relatively simple to reconstruct, and the results in the main text are sufficient to illustrate this. The modified attributes displayed here include 'Bushy_Eyebrows', 'Rosy_Cheeks', 'Mustache'.

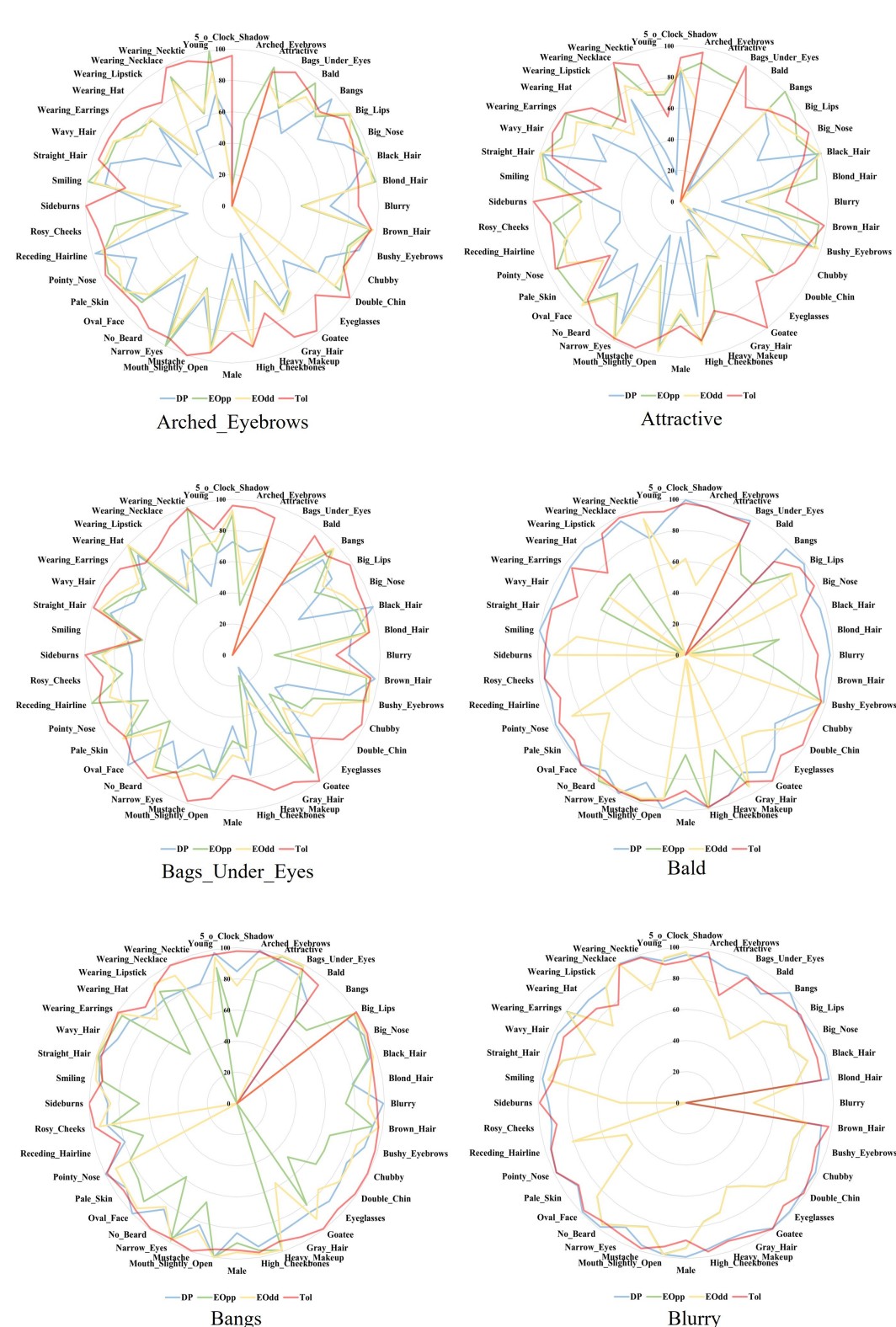

Figure 10: The correlation between different task attributes and the other 40 attributes.

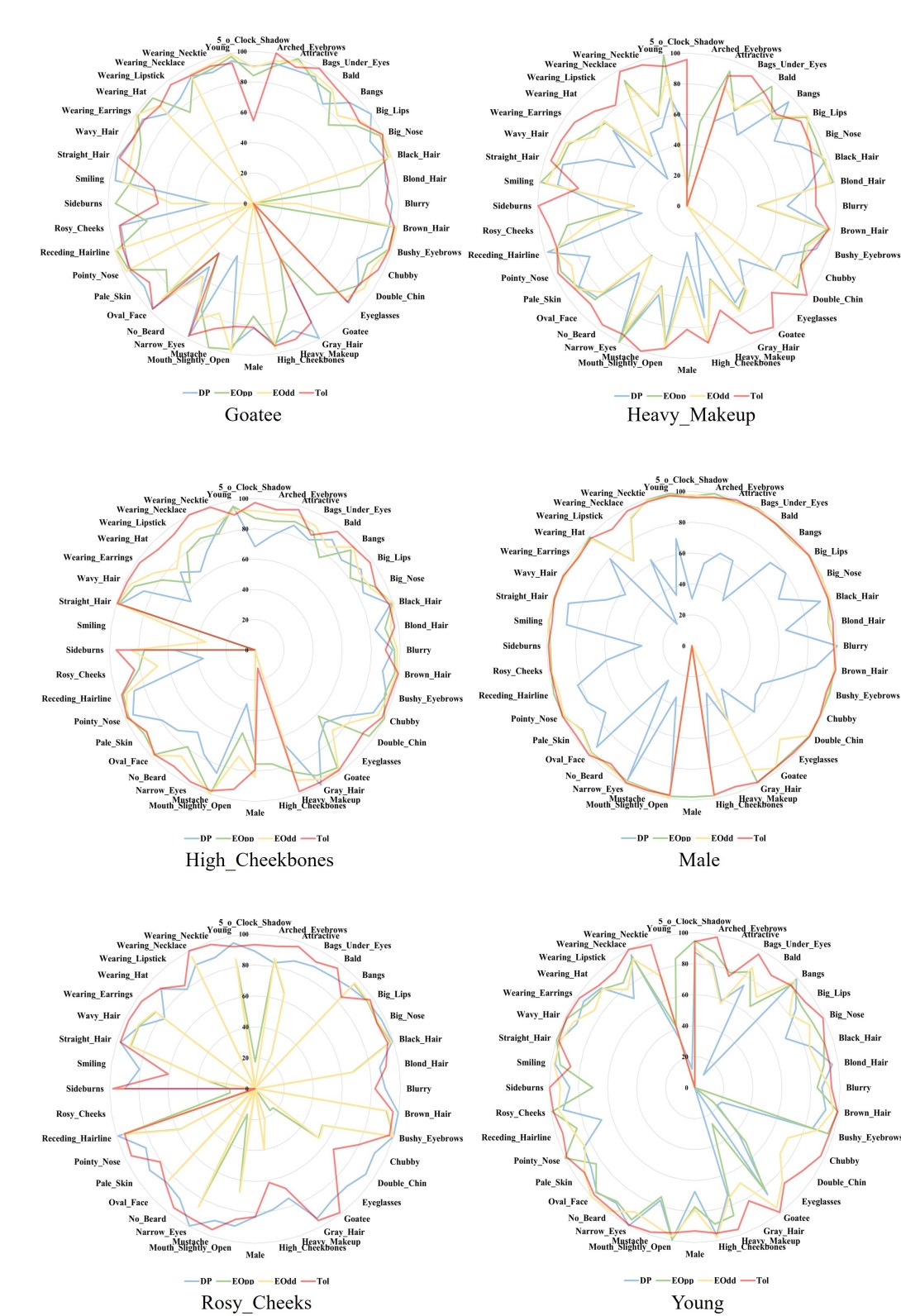

Figure 11: The tab:correlation between different task attributes and the other 40 attributes.

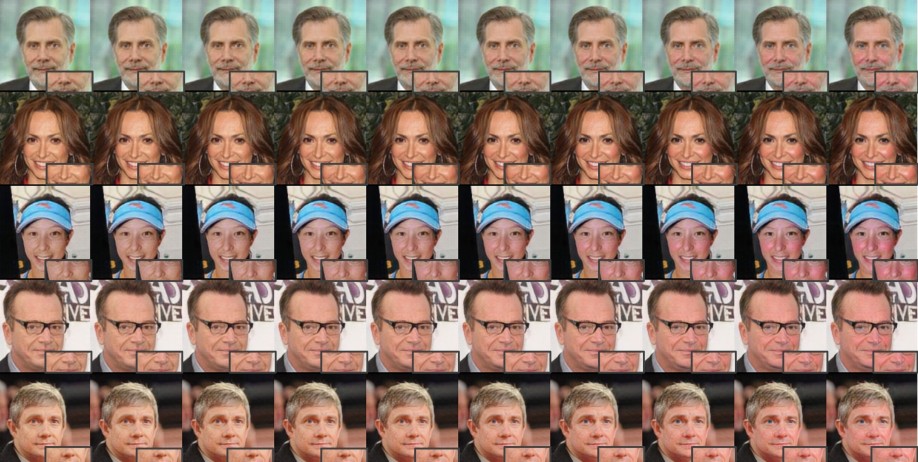

Bushy_Eyebrows

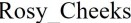

Rosy_Cheeks

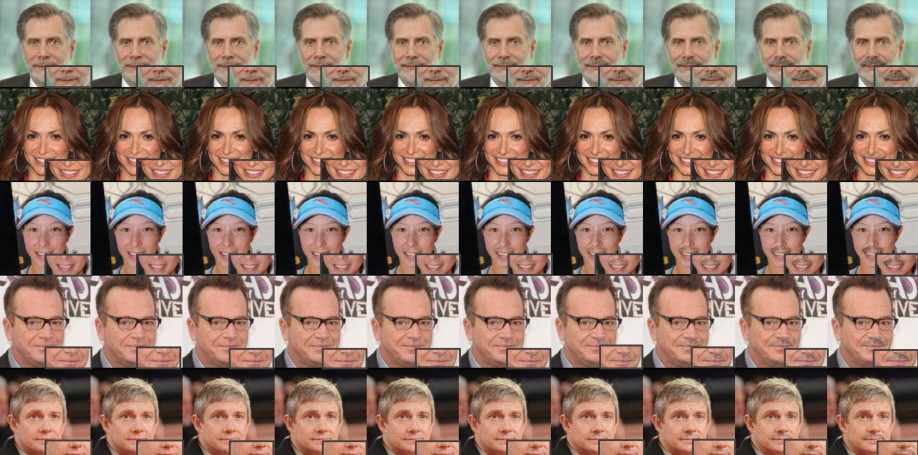

Mustache

Figure 12: The results of dynamic perturbation of attributes in the facial recognition scenario of the CelebA dataset. We visualized the reconstruction results of controlling single attributes: 'Bushy_Eyebrows', 'Rosy_Cheeks', and 'Mustache', based on the projection factor $z$.

