# OpenReview forum: "Is the Fairness Metric Truly Fair?"
_ICLR.cc/2025/Conference — ICLR 2025 Conference Withdrawn Submission_

### Official Review · Reviewer_dkZP · 2024-10-21

**Soundness:** 2
**Presentation:** 3
**Contribution:** 1
**Rating:** 3
**Confidence:** 4

**Summary:**

The paper proposes a method for objectively evaluating fairness in machine learning models, which tries to address the challenges posed by various factors such as imbalanced test data distributions, label noise, and annotator bias, which can introduce subjective biases into the evaluation process. The authors propose a novel evaluation framework for fairness that leverages attribute sampling continuity and attribute correlation independence through dynamic perturbations of input data. Within this framework, fairness is defined in terms of the minimal perturbation required to alter a model's prediction. Based on this definition, the study introduces several new fairness metrics that are defined based on the measurements of the stability of a model's predictions under controlled perturbations.

**Strengths:**

1. The authors study a crucial issue in fairness: the lack of quantitative methods for objective fairness assessment, necessitating reasoning in terms of trade-offs between multiple (conflicting) notions.
2. The concept (underpinning the paper) of studying fairness through the lens of input perturbation is intriguing.
3. The quality of writing is good, well-written paper.
4. Code is provided for reproducibility.

**Weaknesses:**

1. The paper's claim of contribution regarding conflicts in existing group fairness metrics is not novel, as this is already well-documented in previous research [1,2]. Furthermore, current literature suggests that statistical fairness measures lead to inefficiencies in accuracy-fairness trade-offs, advocating instead for min-max fairness evaluation [2].
2. The concept of using generative models to perturb visual attributes for fairness measurement is not original to this work [3,4].
3. A minor concern arises from the attempt to perturb one dimension at a time in the mapped projection factor z, as arbitrary generative models may not effectively decouple input attributes without specific hypotheses on inductive biases [5].
While Figure 4 suggests the mappers perform adequately, this concern should be thoroughly addressed to ensure a robust framework for objective evaluation.


[1] Verma, Sahil, and Julia Rubin. "Fairness definitions explained." Proceedings of the international workshop on software fairness. 2018.

[2] Zietlow, Dominik, et al. "Leveling down in computer vision: Pareto inefficiencies in fair deep classifiers." Proceedings of the IEEE/CVF Conference on Computer Vision and Pattern Recognition. 2022.

[3] Ramaswamy, Vikram V., Sunnie SY Kim, and Olga Russakovsky. "Fair attribute classification through latent space de-biasing." Proceedings of the IEEE/CVF conference on computer vision and pattern recognition. 2021.

[4] Balakrishnan, Guha, et al. "Towards causal benchmarking of bias in face analysis algorithms." Deep Learning-Based Face Analytics (2021): 327-359.

[5] Locatello, Francesco, et al. "Challenging common assumptions in the unsupervised learning of disentangled representations." international conference on machine learning. PMLR, 2019.

**Questions:**

- Given that the min-max fairness setting [1] addresses inefficiencies of statistical parity measures, when evaluating fairness, it would be beneficial for the authors to elucidate the trade-offs between improving their proposed fairness metrics and the overall model performance on the downstream task.


[1] Zietlow, Dominik, et al. "Leveling down in computer vision: Pareto inefficiencies in fair deep classifiers." Proceedings of the IEEE/CVF Conference on Computer Vision and Pattern Recognition. 2022.


_**Justification of rating.**_
I would thank the authors for having studied this subject, as in fairness there's an urgent necessity to rethink evaluation criterias. Although the concepts are interesting and well presented, personally, I don't feel the proposed solution is able to advance the field. Specifically, the fundamental flaws of the fairness metrics currently used is very well known in literature (so I don't see it as a contribution of the paper. See Weakness #1). Furthermore, the evaluation scheme seems an adaptation of methods that have been already tried in literature (Weakness #2) and that have still some open points, for which the paper doesn't propose any solution (see Weakness #3)

---

### Official Review · Reviewer_TSjz · 2024-11-03

**Soundness:** 1
**Presentation:** 3
**Contribution:** 2
**Rating:** 1
**Confidence:** 3

**Summary:**

This paper presents a fairness evaluation system for computer vision models that aims to correct several characteristics in fairness metrics including inconsistencies between metrics and inconsistent scores across sensitive attributes.  This is achieved by projecting images to a latent space designed to map sensitive attributes onto a similar dimension for evaluation providing consistent evaluations across sensitive attributes.  The authors demonstrated existing characteristics of fairness evaluation metrics such as metrics being conflicting or confusing to use, that different sensitive attributes have different fairness scores, and that errors or imbalances in the test attributes can result in inaccurate results.  Following this they demonstrate that their fairness system and the two metrics they propose, Tol and Dev, provide more consistent results across sensitive attributes and tasks.

**Strengths:**

The projection of image data to a latent space for evaluation is an interesting idea that could be used to decouple intersectional sensitive attributes in images for evaluation.  If the goal is to evaluate individual sensitive attributes it is difficult to do this when images such as those used by the authors have many attribute labels per image.  In addition the metrics being based on the degree of perturbation you need to make to an attribute to cause the model to flip it’s classification is a good way to adversarially find the limits of robustness to that attribute.

The authors analyse many different models and compare against a variety of fairness metrics.  They not only highlight the known problems associated with existing fairness metrics but also run experiments to verify those claims.

It was good to see the authors analyse both objective and subjective tasks and refrain from end tasks such as gender classification that is an internal state.

**Weaknesses:**

The primary weakness in this work stems from the claims that different fairness scores across sensitive attributes is a problem, and that fairness metrics are inconsistent.  These are expected and by design.  Fairness is a socio-technical issue where the discovery of biases is just one problem that needs to be considered along with the sensitive attributes that are important as well as the fairness outcome that is desired.  It is completely acceptable that one fairness metric might discover that there is bias against one demographic and not against another, and that these results differ across fairness metrics as they are measuring different types of biases.

Based on the above claims there was too much language in the paper claiming unfair fairness metrics, when in fact the metrics are just limited, but not unfair.

This paper might be better framed as a sensitive analysis method for fairness by decoupling intersectional labeled attributes (if they can indeed decouple the attributes - it seemed that Figure 7 showed that they were not in fact decoupled).

On line 88 the authors say they have “better results” because they are closer to human cognition and more stable across different tasks.  I don’t see in the paper how they show it is closer to human cognition.  There is an experiment with 10 experts (please define experts) but I didn’t understand the description on line 521 onwards that demonstrated this point.  To do this you would need to present data that showed there was a statistically significant result and that you had a sufficient number of subjects to achieve this.

There are some works that do a fine grained analysis on computer vision tasks that are missed in the claim that the field of computer lack them on line 148.  For example:

Luo, Jinqi, et al. "Zero-shot model diagnosis." Proceedings of the IEEE/CVF Conference on Computer Vision and Pattern Recognition. 2023.

Joshi, Aparna R., et al. "Fair SA: Sensitivity analysis for fairness in face recognition." Algorithmic fairness through the lens of causality and robustness workshop. PMLR, 2022.

The title of the paper doesn’t seem fitting to the work in the paper.  This work doesn’t analyse the fairness of fairness metrics or propose a methodology for doing that, which would inevitably involve an analysis of the effects a fairness metric would have on the demographics analysed in an end-task model.  Rather, the authors highlight known characteristics of fairness metrics and propose a sensitivity analysis based technique for overcoming them.

Some of the claims around what is bad about fairness metrics made by the authors are just well-known characteristics of fairness metrics (as the authors pointed out by the relevant citations).  Based on the problem you are hoping to solve in a fairness evaluation you choose the right fairness metric for your problem.  In addition, the claim that different sensitive attributes having different scores is an issue is incorrect.  The goal in a fairness evaluation is to see if there is a fairness issue across a sensitive attribute for an end task.  It is completely normal to expect that for some sensitive attributes there is a bias and for others there is not.  I may have misunderstood the motivation, but I don’t see why you would want to devise a methodology to provide consistent fairness scores across sensitive attributes.

**Questions:**

On line 154, what do you mean by multi-attribute results?

Lines 156 and 158, you say 3 metrics and then 4 metrics.  I don’t believe you used AOdds so it should be 3 right?

Figure 3 (right) is a little confusing.  I didn’t really understand the individual/group/entirety classification of attributes.

Table 1 has lots of duplicated data points in the eyeglasses section.  Are these results correct?  Also the results seem very similar across all models.  Why is that?

How did you get the images in Figure 4?  Figure 7 seems to imply that the latent space doesn’t decouple sensitive attributes.

Line 188 - How was 0-1 reversal used in this context and why?

Line 203 - do you mean that the DP and EOpp lines are overlapping (not DP and EOdd)?

Lines 211-215 - Fairness is a function of the end-task and the sensitive attributes under investigation.  Spurious correlations and imbalanced data can lead many interesting and unintuitive findings.  I’m not sure you can say that the end task of classifying “Bald” which has biases for the sensitive attributes of ‘Mouth Slightly Open’, ‘5 o Clock Shadow’ and ‘Eyeglasses’ is in-contradiction to human cognition and therefore wrong, and then that selecting incorrect or too many sensitive attributes is wrong.  The analysis is to give insight into issue that may exist in the model for an end task and the more explainable data you have for this the more likely it is that you catch a problem and can fix it.

Line 247 - balancing across a single attribute could help mitigate bias for that attribute.  What were the fairness results for these attributes that you balanced?  What about trying to obtain intersectional balance across attributes of interest?

Line 248 - saying that the metrics are inherently unfair due to issues in other parts of the pipeline doesn’t seem right.  The metrics are objective.  How are the metrics unfair?

Line 253 - “FAIENESS” spelling mistake.

Line 256 - the fairness metrics are not ambiguously defined.  They may be difficult to interpret and understand how to use one over the other.  But they aren’t ambiguous.  I also don’t understand why instability across attributes is a bad thing.  It is to be expected.

Line 316 - I don’t understand how the mapped latent space correlates to the sensitive attributes in the labels for the fairness evaluation here.  If it doesn’t this space is uninterpretable and won’t offer an explanation as to why fairness issues arise. You can measure the sensitivity of the model to dimensions in the latent space, but I don’t see how this correlates to fairness.  This feels more like an adversarial attack vector.

Line 334 - This is not the same for a^-.  Presumably you subject sigma in line 7 of the algorithm the second time through.

Line 349 - Is each dimension of the latent space an attribute?

Line 361 - I don’t understand how this allows the “similarity of attributes to be quantified”.  Could you please explain further?

In Table 1, are the fairness scores the average across all sensitive attributes?

Line 428 - Does this mean that your a^+ and a^- search doesn’t find a point where the labels change?  If so what are you measuring then when this happens?  How often does this happen?

Line 431 - How do you normalise the computed metrics.

Line 477 - I’m not sure this is a fair comparison between Tol and Dev as mathematically they are very similar.  It’s not surprising that they are highly correlated.

Line 483-485 - I don’t understand this.  How are you measuring correlation?  How do you know that there isn’t a bias for chubby/double-chin attributes for the task moustache?

Line 500 - These are factors that could help a classifier but they may not be groups with a bias.

Line 511 - “EXTAND” is mis-spelt.

---

### Official Review · Reviewer_TwKV · 2024-11-08

**Soundness:** 1
**Presentation:** 1
**Contribution:** 2
**Rating:** 3
**Confidence:** 3

**Summary:**

This paper claims that current metrics proposed for measuring the fairness of the vision classifiers have inherent flaws. First, they perform an experiment to show how different metrics measure the fairness for various classifier structures. Then, based on this experiment they propose three major observations (which I believe are not supported properly), and then they propose a metric in the feature space to measure the correlation between sensitive and target attributes.

This paper is really hard to follow, lots of concepts are not defined or clarified properly, most of the claims or conclusions lack experimental support, and in it's current form is far from being a good fit for a top-tier conference.

**Strengths:**

The idea of using feature space to measure fairness is interesting.

**Weaknesses:**

This paper has some critical issues that make it hard to understand the underlying idea and evaluate the idea properly:

**1. Writing needs to be improved drastically:**

- Some critical information and required preliminaries are missing. For example:
i) what is the definition of the sensitive attribute, target attribute, and task attribute that is used throughout the paper? (this information needs to be included to make the paper self-contained). A proper preliminary section can clarify this.
ii) The experimental setup in 3.1 is not defined properly.
iii) The setup for 2(b) is not clear to me!! How have you trained the model with the task attribute 'Bald"? Is it supposed to be fair w.r.t. 'Bald'? if yes, how does this affect other attributes like 'Wavy_Hair'?

- Some choices of words make the writing complicated. For example: "... a perturbation space projection factor z ..." (line 309). This is not the best way to express your thoughts and sometimes these types of writing create a huge burden for the reader to understand the idea!

- There are some grammatical errors, even on the headers, for example: Section 4 header $\rightarrow$ "Faieness".


**2. Section 3 (the analysis of the previous metrics) has several issues:**

- One major limitation of this analysis is that the problem setup, the training details, and the objective are not clear to the reader.

- How do different architectures result in so much difference in the measurement?  For example, how does moving from RN-18 to RN-34 affect the results that much?

- This is more like considering the effect of the network structure on the metric rather than evaluating the metric itself.
Analyzing the metrics needs a deeper evaluation of how the metric is sub-optimal in capturing a specific fairness behavior!



**3. Most of the claims in the paper have not been supported properly with either experimental results or findings in previous works, and these are making the foundation of the proposed method:**

- I can not understand how  this claim *"Subjectivity in labeling and imbalance in test attributes result in inaccurate results."* (line 231) is related to the results in Figure 1 (despite reading it several times).

- (line 233) *"First, the subjectivity inherent in data annotation is itself a significant source of unfairness in computer vision
models, primarily manifesting as annotator bias (Sheng et al., 2019; Berinsky et al., 2012), ambiguous
labeling standards (Snow et al., 2008; Hovy et al., 2016), and cultural bias (Geva et al., 2019;
Sap et al., 2019)."* How does your approach address this issue? If not, why this is discussed in your observations?

- The opening of section 4 has several issues and is not strong enough. For example:
i) what is the ambiguity about the definitions of the metrics? (line 256)
ii) how imbalanced data could be the problem of the metric? this problem occurs because we do not have a properly balanced dataset, and not that there is an intrinsic issue with the metric! (line 257)
iii) I do not understand how this issue can lead to a subjective bias.  (line 258)

- The opening of section 5 (line 298) discusses some conclusions from section 4 (Eq. 1 which only defines a projection space and then the minimum shift to flip the label) as follows: *"We revisited the perspective of fairness evaluation and introduced a projection fairness evaluation framework for visual deep models, achieving unified mapping of various sensitive attributes and their values, and approximating the fairness boundary to evaluate the fairness of image classification models at different levels."*
**I am not sure how this is achieved in section 4 or earlier sections!!**

- (line 311) *" One potential solution is to use the disentanglement capabilities of autoregressive or generative models as mappers, generating a more comprehensive latent code that covers sensitive attributes within the dataset.*
How exactly is this disentanglement capability related to the goal of this section? How we can evaluate that this serves our purpose? Is there any control mechanism for this? Given that for most of the generative models, usually, we have no exact control and learn the encoded knowledge in z space by data.

- (line 320). This is also somehow related to the disentanglement concept discussed. How we can make sure that this d-th dimension in z space is related to a meaningful attribute?
We need to have a sensible example for this and enough support to show that this is related to a sensitive attribute.
For example, if we have a certain classification task, and for this task Gender is a sensitive attribute, we need to show that in fact on of these dimensions in z space encodes Gender information.

**4. Multiple issues mentioned in the analysis and proposed method make it hard to evaluate the experimental results.**

**Questions:**

Please refer to weaknesses.

---

### Note · Authors · 2024-11-22

I have read and agree with the venue's withdrawal policy on behalf of myself and my co-authors.